



**Variability of Relativistic Electron Flux (E>2 MeV) during Geo-Magnetically**
**Quiet and Disturbed days: A Case Study**
**Tulsi Thapa[1,2], Binod Adhikari[1,3], Prashrit Baruwal[4,] Kiran Pudasainee[1]**
[1]Department of Physics, St. Xavier's College, Maitighar, Kathmandu, Nepal
[2]National Astronomical Observatories of China, University of Chinese Academy of Sciences,
China
[3]Department of Physics, Patan Multiple Campus, Tribhuvan University, Kathmandu, Nepal
[4]Central Department of Physics, University Campus, Tribhuvan University, Kirtipur, Nepal
*Corresponding Author: binod.adhi@gmail.com*
**Abstract**
We analyzed the relativistic electron fluxes (E>2MeV) during three different geomagnetic
storms: moderate, intense, and super-intense and one geo-magnetically quiet period. We have
opted Continuous wavelet analysis and cross-correlation technique to extend current
understanding and of the radiation-belt dynamics. We found that the fluctuation of relativistic
electron fluxes dependent basically on prolonged southward interplanetary magnetic field IMF-
Bz. Cross-correlation analysis depicted that SYM-H does not show a strong connection either
with relativistic electron enhancement events or persistent depletion events. Our result supports
the fact that geomagnetic storms are not a primary factor that pumps up the radiation belt. In fact
they seem event specific; either depletion or enhancement or slight effect on the outer radiation
belt might be observed depending on the event. Solar wind pressure and velocity were found to
be highly and positively correlated with relativistic electron. We found that, the count of
relativistic electron flux (> 2 MeV) decreases during the main phase of geomagnetic storm with
the increase in -- from quiet to super intense storm -- geomagnetic storm conditions (Table 1).
However, Psw was found to be weakly correlated in case of intense storms following an abrupt
increase of electron flux for ~4 hrs, which is interesting and unique.

**Keywords**:Geomagnetic Storms, Relativistic Electron, Cross-Correlation, Continuous Wavelet
Transform





## 1. Introduction


The major plasma sources in the interplanetary medium responsible for geomagnetic
disturbances are identified as coronal mass ejection (CMEs), which include the magnetic cloud,
interplanetary shock, the co-rotating interaction region (CIR) and the high-speed solar wind
streamers[Gosling et al., 1991]. The interaction between those interplanetary structures with the
Earth's magnetosphere-ionosphere system can produce effects such as geomagnetic storms, sub
storms and trapping of high energy charge particles in the radiation belt, known as Van Allen
belt[Mauk et al., 2012]. There are various solar wind parameters that are effective enough to
fluctuate the content of relativistic electron flux. Enhancement in relativistic electron fluxesmight
be an important sources of energy input and chemical change to the middle atmosphere.
Magnetic reconnection is the main physical phenomena transporting energy from the solar wind
into the magnetosphere.

Van Allen radiation belts are composed of ions, protons and electrons with energy ranging from
100 k eV to 10 MeV. It consists of two belts: inner and outer radiation belt. Outer radiation belt
usually lies at an altitude of 3 Earth radii ($R_E$)  and extending to 10 $R_E$ above the Earth's surface
where GPS satellites, metrological satellites, broadcasting and communication satellites are
operating [R. Kataoka and Y. Miyoshi, 2008]. The increasing dependency on modern
infrastructure and technology and expanding human presence in space drags us more for the
comprehensive study and understanding of space weather and their dynamics [Baker et al.,
2000]. For the deeper understanding of the structure and dynamics of Earth's radiation belt,
NASA developed the Van Allen Probes mission [Mauk et al., 2012].The aftermath of highly
fluctuating electron fluxes in the Earth's outer radiation belt might be its acceleration and loss.
Paulikas & Blake [1979] reported such rapid acceleration and loss of relativistic electrons.
Reeves et al., [2003] and Turner et al. [2013]   added relativistic electron population in the
radiation belt can not only subsidize but also can be enhanced, depleted, or even not affected at
all due to the acceleration and loss mechanism. Pinto et al. [2018] identified and analyzed 61
relativistic electron enhancement events and 21 depletion events during 1996 to 2006, resulting
the persistent depletion events are characterized by: a low Vsw, a sudden increase in proton
density, and a northward turning of IMF Bz. Also,predicted their threshold values.





This work focuses on the loss, acceleration, and transport of relativistic electron E >
2 *MeV* during three different geomagnetic storms:moderate, intense, and super-intense and one
geo-magnetically quiet period.

## 2. Dataset and Methodologies

The datasets has been extracted from OMNI (Operating Mission as Nodes on the Internet)
webpage and downloaded from the official website of NASA
https://omniweb.gsfc.nasa.gov/dataset to study solar wind parameters. The integrated fluxes of
electrons with energies E > 2 *MeV* at geosynchronous orbit ($L = 6.6$) for our study are collected
from the GOES-8 and GOES-12; *Geostationary Operational Environment Satellites*
(*GOES*; http://www.ngdc.noaa.gov/stp/satellite/goes/dataaccess.html). This database provides 1
min temporal resolution data obtained from the sets of ACE, Wind, and IMP-8 satellites.The
wavelet transform, particularly continuous wavelet transforms, which helps to understand the
behavior of the energy at different scales and the cross-correlation techniques to find the relation
between the relativistic electron and different parameters of the solar wind have been
implemented. The detailed explanation of the theory associated can be found in various research
papers [eg. Adhikari et al., 2015,2017(a), 2018; Usoro A. E., 2015].

### 2.1 Continuous Wavelet Transform

Wavelet transform is an effective mathematical tool for the analysis of transient signals. A
continuous wavelet transform (CWT) maps a one-dimensional signal to a two-dimensional time-
scale that produces a time- frequency decomposition of the signal which segregates individual
signal components constructively unlike the short time Fourier transforms (STFT)., The square
modulus of the wavelet coefficient, in analogous to the Fourier analysis, lays out the energy
distribution in the time-scale plane [Adhikari et al., 2018].In our study, using CWT scaleogram,
the vertical axis provides the information of periodicity at different scales as a function of time in
horizontal axis. It helps to comprehend the behavior and distribution of the energy at different
scales [Adhikari et al., 2017b]. The detailed analysis of CWT are shown in Figure 5.

### 2.2 Cross-correlation





Cross-correlation is the standard, multi-time scale, statistical tool that helps out to obtain the
time-delay, determine the similarities, draw similar relative characters and can furnish with the
new information [Adhikari et al., 2018]. The time scale is used to determine the lead or lags
between the parameters after establishing their correlation. The horizontal plane includes the
time (minutes) ranging different values and vertical plane indicates cross-correlation coefficient
[Adhikari et al., 2017a]. The detailed analysis of different events are shown in Figure 6.
We have taken four different events as listed in table 1.

| Events | Year/Month/day | SYM-Hvalue(nT) | Event type |
|---|---|---|---|
| **Event 1** | 2007/01/25 | 0 to -50 | Quiet period |
| **Event 2** | 2008/09/04 | -50 to ≥-100 | Moderate |
| **Event 3** | 2006/12/15 | -100 to ≥ -250 | Intense |
| **Event 4** | 2001/03/31 | ≤-250 | Super intense |

**Table 1**: Selected storm events, their occurrence time frame in year, month, and day, SYM-H value (nT), and event types are listed out.

## 3. Results and Discussion

The relativistic electron population in the outer radiation belt is extremely volatile during periods
of enhanced geomagnetic activity. Electron fluxes are commonly seen to be depleted during the
storm main phase, fluxes can recover to pre-storm levels in the recovery phase and stay depleted
or build up to exceed pre-storm levels [Reeves et al., 2003; Meredith et al., 2011]. The flux
dropout can be a combination of adiabatic and non-adiabatic effects and losses through the
magnetopause and atmospheric precipitation [Millan and Thorne, 2007]. In this work, we have
selected four different events depicted in Table 1. The selection of storm types is based on the
SYM-H value in accordance with the explanation by Perreault et al.,[ 1978];Gonzalez et al.,
[1994]; and Wanliss et al.,[ 2006].

**3.1.1 Event 1: The Quietest day (January 25, 2007)**



Figure 1 depicts the quietest day, 25th January 2007. The value of SYM-H, on sixth panel, falls
to minimum of -30 nT at ~20:00 UT and at the same time IMF-Bz~-4. In fact, the fluctuation of
IMF-Bz is mostly southward for almost the entire day with small variations, may be due to the
presence of Alfven waves [Adhikari et al.,2015].The first panel at the top of the Figure 1
represents the fluctuation of solar wind pressure (~2 nPa to ~2.5 nPa). The second panel shows
the fluctuation of solar wind (Vsw) (~750km/s at 00:00 UT and gradually decreases to its lowest
value of ~600 km/s towards the end of the day). The fluctuation of solar plasma density (Nsw) is
represented by fourth panel (value lies between ~2 to ~3 n/cc). As the solar wind pressure is not
so high enough to compress the magnetosphere, the high speed solar wind will bring the
energetic solar wind plasma into the magnetosphere. Since, the IMF-Bz component is mostly
southward through the day with little fluctuation, allowing the energetic particles to inject into
the magnetosphere. Thus, the relativistic electron fluxes (E>2 MeV) seems to be populated in the
radiation belt showing maximum value of ~3500 $cm^{-2}S^{-1}Sr^{-1}$, here after we call flux unit
as'FU' for convenience, with slight fluctuation recording its lowest value of ~2700 FU at 22:00
UT (shown in the fifth panel of Figure 1). As high speed solar wind streams come in contact with
the magnetosphere, the electrons gain their acceleration [Baker et al., 1993; Paulikas et al.,
1978]. The bottom panel indicates the fluctuation of AE index, reaching maximum of ~800 nT
corresponding to the minimum value of SYM-H.




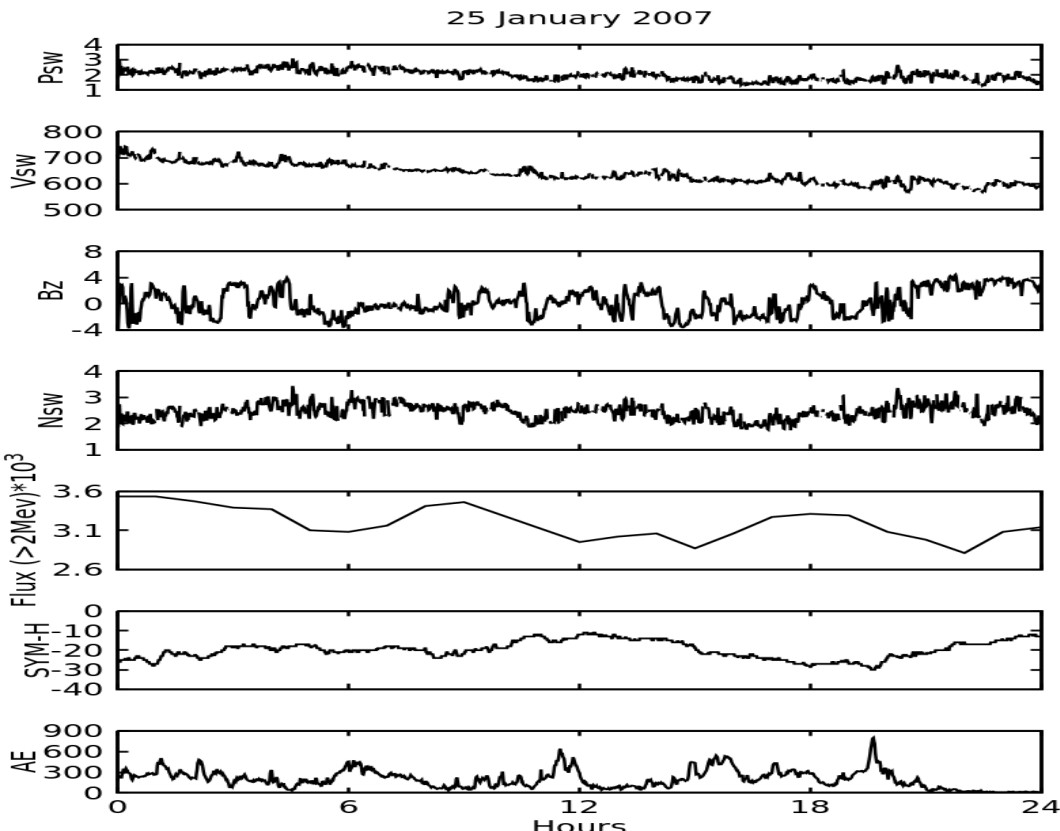

**Figure 1**: From top to bottom, the panels show the variations of: solar wind pressure (Psw in
nPa), solar wind speed (Vsw in km/s), interplanetary magnetic field (Bz in nT),  solar wind
plasma density(Nsw in n/cc), relativistic electron flux $E > 2\,MeV$(Flux in $cm^{-2}S^{-1}Sr^{-1}$),
symmetric horizontal component of magnetic field (SYM-H in nT), and auroral electrojet (AE in
nT) indices with time (Hours) respectively for event-1 of  25 January 2007.

**3.1.2 Event 2: Moderate Storm (September 04, 2008)**
Figure 2 shows the fluctuations in different interplanetary structures during a moderate storm
occurred on September 04, 2008. The sequence of panel is same as that explained in previous
event. The value of SYM-H drops down to ~-70 nT indicating the storm as moderate storm as
defined by [Perreault et al., 1978;Wanliss et al., 2006]. The fluctuation of IMF Bz(in third
panel), directed southward during storm main phase, allowing the charge particles to enter easily
in the magnetosphere[Lemaire, J. F.,2012].The seventh panel shows the fluctuation of AE index

137





with value ~1600 nT corresponding to lowest value of IMF-Bz, indicating the normal auroral
activities. The first panel at the top of figure 2 shows the solar wind pressure around ~9nPa at the
early phase of the storm at 01:00 UT and gradually goes on decreasing to attain a lowest value
of~1 nPa at the end of the day. The fluctuation of solar wind velocity at the second panel shows
the gradual increment of its magnitude which is about 480 km/s at 00:00 UT and reaching 600
km/s at 19:00 UT. The flux of relativistic electron is almost constant with value ~100 FU
until16:00 UT and then starts to accelerate to maximum value of ~2900 FU at 24:00 UT.
Rothwell and McIlwain [1960] formulated a general pattern of the relativistic electron dynamics
during a geomagnetic storm. On the study of 276 geomagnetic storms, Reeves et al. [2003] found
that only 53% were associated with an enhancement event at geostationary orbit, while 19%
were associated with a net flux loss, and 28% showed no significant change which are in
agreement with other studies [e.g., Moya et al., 2017; Turner et al., 2013; Zhao & Li, 2013],
confirming that solar storms are not always associated with enhancement or depletion in
relativistic electron fluxes. Moreover, WladislawLyatsky and Khazanov [2008] found that both
Vsw and Nsw provide a strong effect on relativistic electrons. Furthermore, Reeves et al., [2011]
have shown that relativistic electron enhancements are correlated (however, not linearly) with
the presence of high-speed solar wind steams and southward IMF during the recovery phase of a
geomagnetic storms. Higher the solar wind velocity, higher would be the probability of energetic
electron buildup after a storm activity (the recovery phase); during which the evolution of sub
storm activity may be due to the gradual enhancement of relativistic electron [Kilpua et
al.,2015].Anderson et al. [2015] showed that the impact of acceleration and loss of relativistic
electrons on radiation belt seems to be negligible in comparison to both small and large
geomagnetic storms.





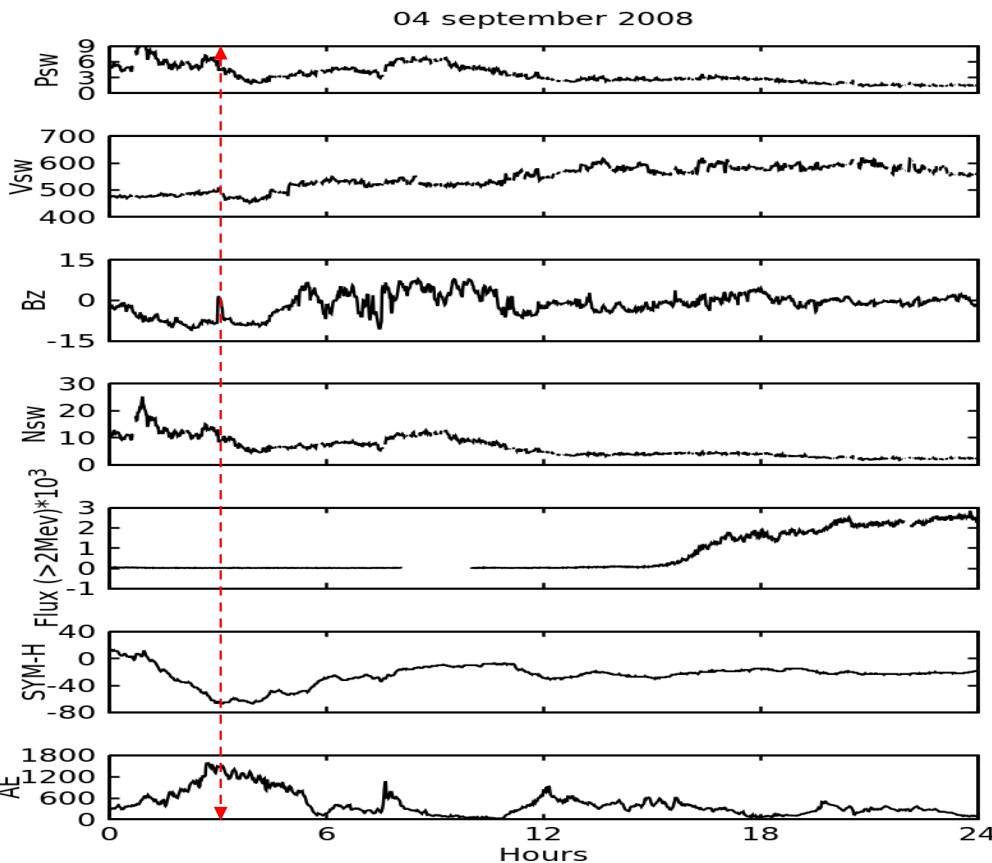


**Figure 2**: From top to bottom, the panels show the variations of: solar wind pressure (Psw in nPa), solar wind speed (Vsw in km/s), interplanetary magnetic field (Bz in nT), solar wind plasma density(Nsw in n/cc), relativistic electron flux  E$> 2\ MeV$(Flux in $cm^{-2}S^{-1}Sr^{-1}$), symmetric horizontal component of magnetic field (SYM-H in nT), and auroral electrojet (AE in nT) indices with time (Hours) respectively for event-2 of 04 September,2008.


**3.1.3 Event 3: The Intense Geomagnetic storm (December 15, 2006)**
Figure 3 shows the fluctuation of solar wind parameters, component of interplanetary magnetic
field Bz, flux of relativistic electrons (E $> 2\ MeV$)geomagnetic indices SYM-H, and AE during
an event on December15, 2006. The fluctuation pattern of these parameters is as same as that
defined in previous events. In the fifth panel of Figure 3, SYM-H shows maximum disturbance
of about ~-220 nT at 01:00 UT and stays under -100 nT until 12:00 UT, indicating the storm





condition as intense geomagnetic storm as categorized by [Perreault et al., 1978; Wanliss et al.,
2006].In the last (seventh) panel the fluctuation of AE index is shown which reaching its value of
around ~1200 nT corresponding to which the IMF Bz (shown in third panel) shows~-18nT
continuing to recover towards northern direction. In the fifth panel, we have the fluctuation of
relativistic electron flux which is almost constant during the storm time attaining the value of
about ~1100 FU until 12:00 UT and abruptly accelerating to maximum of ~46000 FU at 14:00
UT for almost an hour and steeping towards normal. The high solar wind speed (reaching the
value of ~850 km/s at 01:00 UT, as shown in second panel) meant the injection of larger amount
of relativistic electron flux inside the radiation belt. Instead, the fluxes were seen to be depleted
and rather increasing its value for few hours as the velocity decreases to ~600km/s during the
recovery phase. Reeves et al., [2003] analyzed the data set that compared 15 years of solar wind
data of MeV electrons flux, resulting the high solar wind speeds not as the primary factor for
enhancement of relativistic electrons; instead, increment in southward IMF value is the essential
condition to cause acceleration of MeV electrons in the outer radiation belt. In addition,
Borovsky and Denton, [2005] found that for high-speed stream-driven storms, there is
considerable spatial overlap of the super-dense ion plasma sheet with plasmaspheric drainage
plumes. This would lead to growth of electromagnetic ion-cyclotron waves that can cause
relativistic electron precipitation loss. Since the super dense plasma sheet is associated with high
Nsw, large Nsw would enhance such loss. This seems to be the case for this event as well. The
first panel shows the fluctuation of solar pressure which remains under 4 nPa throughout the
event.



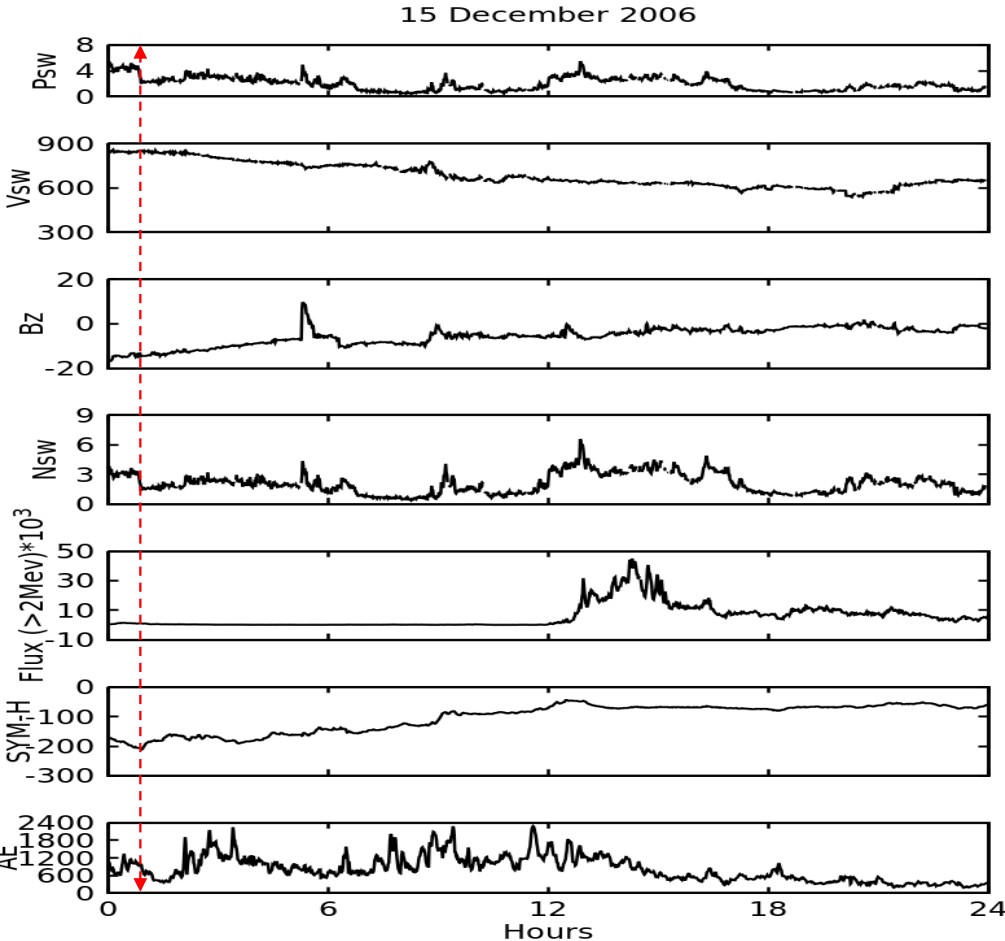

**Figure 3**: From top to bottom, the panels show the variations of: solar wind pressure (Psw in nPa), solar wind speed (Vsw in km/s), interplanetary magnetic field (Bz in nT), solar wind plasma density(Nsw in n/cc), relativistic electron flux $E > 2\ MeV$(Flux in $cm^{-2}S^{-1}Sr^{-1}$), symmetric horizontal component of magnetic field (SYM-H in nT), and auroral electrojet (AE in nT) indices with time (Hours) respectively for event-3 of 15 December 2006.

### 3.1.4 Event 4: The super intense storm (March 31,2001)

In the figure 4, the panels from top to bottom, show solar wind pressure (Psw), solar wind velocity (Vsw), Southward component of magnetic field (IMF-Bz), the solar wind density (Nsw), relativistic electron flux ($E > 2\ MeV$), the geomagnetic indices SYM-H, and AE during



march 31, 2001 respectively. In addition, in the sixth panel, the gradual decay of SYM-H value
up to -410 nT at around 08:00 UT indicates the occurrence of geomagnetic storm as super-
intense [Perreault et al.,1978;Wanliss et al., 2006]. The rapid fluctuation of solar wind
parameters started with the compression of bow shock at around 01:00 UT. The main event
occurred at around 08:00 UT and lasted for several hours. The third panel shows the variation of
IMF-Bz component, having strongly negative value ~-50 nT at 06:00 UT until 08:00 UT
corresponding to the SYM-H value of ~-410 nT indicating that the magnetopause was briefly
pushed inward of geostationary orbit, IMF Bz turned very dynamic, fluctuating between ~-50 nT
and~+50 nT, allowing the charged particles to penetrate repeatedly into the magnetosphere
causing high auroral activity (AE ~2400 nT, shown in seventh panel). During the initial phase of
main storm, the solar wind velocity shows abrupt increment upto ~850 km/s which then
maintains to ~700 km/s during storm main phase. At the same time, the solar wind pressure (first
panel of figure 4)elevates to maximum value of ~60 nPa at the initial phase of the storm and then
decreases to ~20 nPa at 06:00 UT, maintaining same throughout the day. The fluctuation of Nsw
with maximum peak value about ~60 n/cc at 05:00 UT, as shown in fourth panel of figure 4,
followed by the decreases in its value and thereafter low fluctuation is observed. The flux of
relativistic electron, shows maximum value of ~740 FU at 00:00 UT and decreases to ~160 FU
after 01:00 UT. The possible mechanism for this scenario is that for high-speed stream driven
storms, there is a considerable spatial overlap of super dense ion plasma sheet with
plasmaspheric drainage plumes [Borovsky and Denton, 2009]. This would lead to the growth of
electromagnetic ion-cyclotron waves that can cause relativistic electron precipitation loss. As the
super dense plasma sheet is associated with high Nsw, large Nsw would enhance such loss [Li et
al., 2011]. This implies that Nsw clearly plays a crucial role in relativistic electron loss.
However, the strong effect of solar wind density on relativistic electron is relatively unexpected
because the former is not a primary factor in the generation of geomagnetic disturbance [Wanliss
and Showalter, 2006]. Nevertheless, the possible effects of solar wind density on relativistic
electron may be due to the compression of dayside magnetosphere, by high-density solar wind,
acting as shielding in the inner atmosphere thus preventing the penetration of the large-scale
electric fields ultimately leading to losses of relativistic electron form the outer radiation belt.
This indicates the implicit effect of Nsw on relativistic electron.

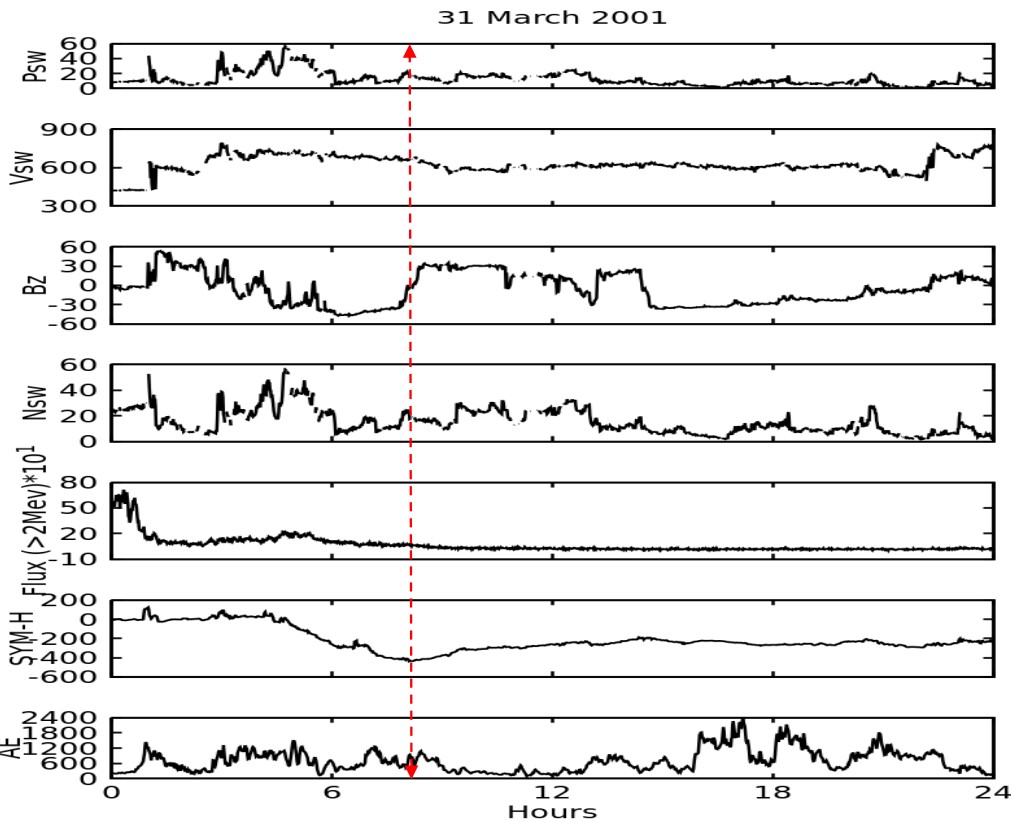


**Figure 4**: From top to bottom, the panels show the variations of: solar wind pressure(Psw in nPa), solar wind speed (Vsw in km/s), interplanetary magnetic field (Bz in nT), solar wind plasma density(Nsw in n/cc), relativistic electron flux $E > 2\,MeV$(Flux in $cm^{-2}S^{-1}Sr^{-1}$), symmetric horizontal component of magnetic field (SYM-H in nT), and auroral electrojet (AE in nT) indices with time (Hours) respectively for event-4 of 31 March 2001.


**3.2 Continuous Wavelet Transform**
Figure 5 depicts the scalograms for the relativistic electrons (E>2 MeV).In Figure 5,magenta
indicates the highest power areas while red represents low power areas value as shown on the
vertical color bar of each plot on the right. Furthermore,Time (in minutes) are placed on the
horizontal axis whereas period (in minutes) in the vertical axis.
Figure 5 (a) represents the scalogram for the main event of intense storm, stronger wavelet power
areas of flux intensities between 0.4-0.7 FU are accumulated around 65-85 minutes at the time





scales (period) range of 14-16 minutes. This shows the flow of energy lasted for few minutes, the
low flux intensities (less intense power area) dominated the event. Figure 5 (b) and (c)
represents the scalogram plot for the main event of super-intense and moderate storm
respectively. Moreover, the wavelet powers areas of higher flux intensities ranging from 1.2 to 2
FU are found to be distributed at lower to higher time scales range of 2-15 minutes throughout
the event time. In Figure 5(b), the highest power areas is around 60-100 minutes at time scales
approximately 1-3minutes and again at time scales approximately 10-15 minutes. The other
power areas covered by blue color are seen scattered at different times throughout the time scales
between 10-170 minutes. Figure 5(c) shows the scalograms for moderate storm where the highest
power area is around 65-85 minutes at time scales approximately 14-16 minutes. The other
power areas covered by blue color are seen around 60-90 minutes at timescales between 12-16
minutes. Furthermore, such variation of flux intensities on both figures resemble the highly
perturbed state during the happening of main event as compared to the quiet event shown in
Figure 5 (d), where low intensities dominated the wavelet powers areas at different times and
scales. The power areas covered by green color are seen distributed discontinuously over the
cone of influence. The non-uniform distribution of energy at different time scales inconsistent
with time strongly suggests the true dynamic nature of relativistic electron flux. This observation
insists that the geomagnetic storm is not the primary cause of accumulation or loss of charge
particles over the radiation belt. In fact, it shows event-specific behavior.





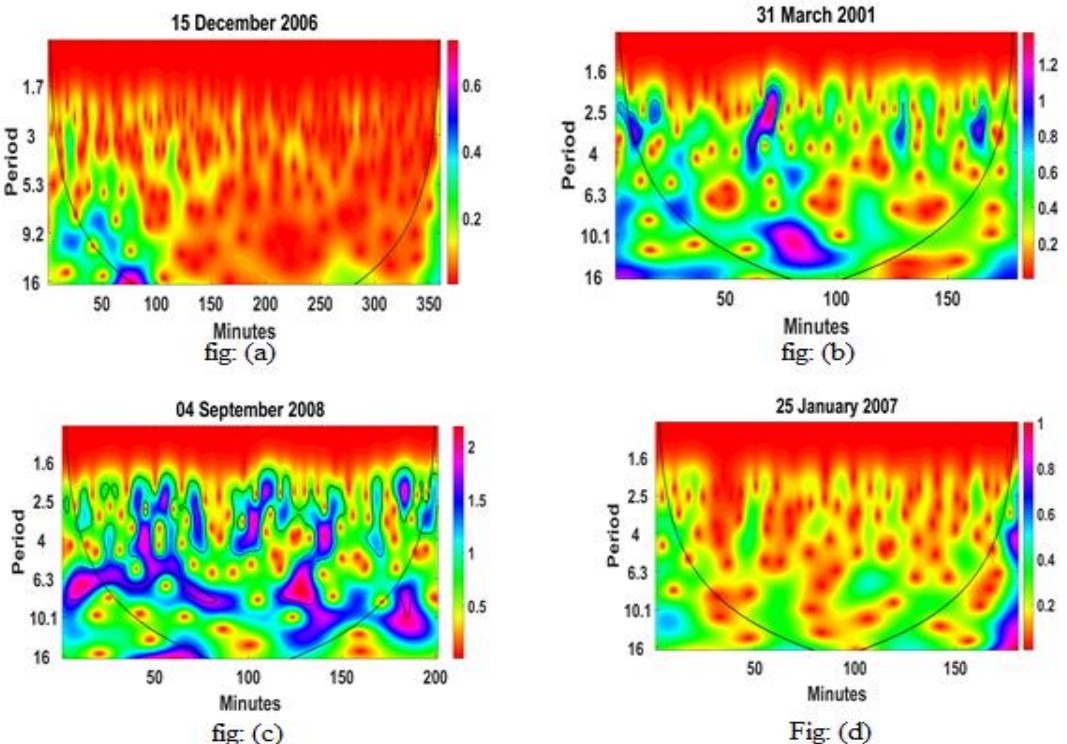

**Figure 5:** Scalograms for the relativistic electron flux (Rel.) E > 2 $MeV$ for (a)Intense storm, 15
December,2006 (b) Super intense storm,31 March,2001 (c)Moderate storm,04 September, 2008,
and (d) Quiet Period, 25 January,2007.

**3.3 Cross Correlation**
Figure 6 shows the cross-correlation between the relativistic electron flux E > 2 $MeV$ (Refer to
as Rel. in legend) with different solar parameters. It is clearly evident that the magenta curve
(Rel.-Nsw), green curve (Rel.-Vsw)and red curve (Rel.-Psw) almost overlap to each other and
attains a good positive correlation with cross-correlation coefficient of 0.9 at lags of 0 minute
(when no lags is applied). In addition, no large variation is observed in the value of cross-
correlation coefficient for relativistic electron flux and solar wind velocity during main phase of
three geomagnetic storm events (our work) and a geoeffectively quiet event. This suggest that
high solar wind velocity is an important condition for enhancing the high radiation belt electron
fluxes but not a determining condition. However, on intense storm, fig (d), the red curve shows
weak correlation with cross-correlation coefficient of 0.3. Indeed, this result is expected as Vsw,





Psw,and Nsw are taken as the plausible factors for the fluctuation of relativistic electron in outer
radiation belt [WladislawLyatsky and Khazanov, 2008]. Li et al. [2005] studied the cross-
correlation of electron flux at 50 Kev to 2 MeV energies with Vsw; the strong correlation with
zero lag for low energies and longer lags for higher energies was observed. Paulikas and Blake
[1979, 2010] found a good correlation between the solar wind velocity and the MeV electron
flux at geostationary orbit.However, they found out the unprecedented complex relationship
between solar wind velocity and radiation belt electron fluxes. It has also been established that
high-speed solar wind and geomagnetic activity are strongly associated with the appearances of
relativistic electrons [Baker et al., 1993]. For the identification of relativistic electrons, the rapid
increment of the sustained solar wind velocity greater than 450 km/s acts as a strong external
indicator [Reeves et al., 2001]. In each of the events, SYM-H seems to be negatively
correlated (in fact it is about 85-90% in average) and so does the IMF Bz. Hence it is not a
reliable predictor as decrease in SYM-H value in not necessarily for relativistic event to occur.

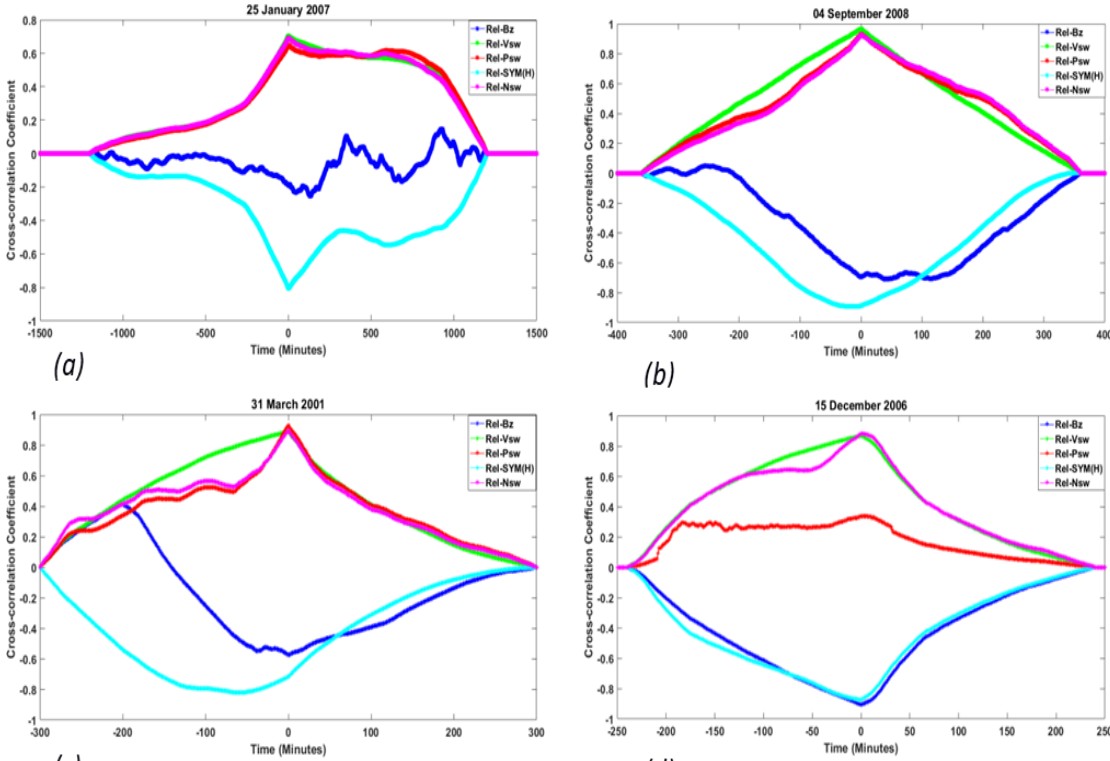






**Figure 6:** Cross correlation between Relativistic electron flux (E>2MeV) with Bz (blue), Vsw (green), Psw (red), SYM-H (sky blue) and Nsw (pink) during (a) Quiet Period, 25 January,2007 (b) Moderate storm,04 September, 2008 (c) Super intense storm,31 March,2001 and (d) Intense storm,15 December,2006 .


**4. Conclusion**
In this paper, we have analyzed various solar wind parameters and geomagnetic indices with
relativistic electron flux ($E > 2\,MeV$) datasets for the four geomagnetic events selected. To
enhance the analysis of relativistic electron flux, the cross-correlation and CWT analysis was
adopted [Adhikari 2015; Adhikari et al. 2017a, b; Adhikari et al., 2018]. The work presented
here shows good correlation, having positive cross-correlation coefficient ($> 0.8$), for
relativistic electron flux with the solar wind velocity, pressure, and solar density. Thus, the
choosing of these parameters are well justified as they are highly geoeffective. In each of the
event, except super-intense, whenever the solar wind exceeds 600 km/s, there seems to increase
in relativistic electron flux. This is consistent with previous results which shows a large average
solar wind speed (Vsw> 500 km/s) is characteristic of enhancement events [Paulikas& Blake
1978, 2010; Reeves et al., 2011].
As depicted in the event-4 (Figure 4), with the increase in solar wind dynamic pressure, the
electron flux shows a decrement factor of 3. The solar wind dynamic pressure and IMF Bz play
indispensable role in causing the relativistic flux dropouts as the magnetopause is compressed
closer to earth or located very far (>10 Earth Radii) [Gao et al., 2015].  The impingement of high
speed solar wind, suppressing the dayside magnetosphere enhancing the drift shell splitting of
charged particles, has impacts onthe possible loss mechanisms of the magnetospheric relativistic
electron, as in the case of super intense storms (super sub-storms). Our result substantiates the
geomagnetic storms are not a primary factor that pumps up the radiation belt. In fact, the
geomagnetic storms can deplete, enhance or cause little effect on the outer radiation belt. To be
precise, it shows event-specific behavior.
In addition, all the geomagnetic storm events, except intense storm, Psw is found to be highly
and positively correlated with relativistic electron flux. The higher Psw leads to the depletion of
the electron flux presumably through compressing the inner magnetosphere and intensifying the





ring current [Gao et al., 2015].However, for intense storm, Psw is found to be weakly correlated
followed by an abrupt increase of electron flux value for ~4 hrs, which is interesting and unique.

In sum, the count of relativistic electron flux (> 2 MeV) decreases during the main phase of
geomagnetic storm with the increase in -- from quiet to super intense storm -- geomagnetic storm
conditions (Table 1). Furthermore, in case of intense geomagnetic storm: during the post-storm
condition, sharp increase in flux count (even larger than in normal quiet condition) is observed,
inkling of unperturbed ionosphere which might be the precursor for any sort of space weather
effects in near future. Thus, there is a need of careful and extensive study of large events over
extended period through more advanced tools and techniques for better understanding of the
inherent physical mechanism.

**Acknowledgements:**
The solar wind, interplanetary magnetic field, geomagnetic indices and relativistic electrons data
for this study were obtained from https://omniweb.gsfc.nasa.gov/. T.Thapa acknowledges the
grant for M.Sc dissertation from the B.P.Koirala Memorial Planetarium.
**Conflict of interest statement**
On behalf of all authors, the corresponding author states that there is no conflict of interest.
**Financial and Ethical disclosures-**
No Funding. Request to waive publication fees.

**Authors should include the following statements (if applicable) in a separate section**
**entitled"Compliance with Ethical Standards" when submitting a paper**
Disclosureof potential conflicts of interest research involving human participants and/or animals
informed consent.






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
