# Peer review of "Variability of Relativistic Electron Flux (E>2 MeV) during Geo-Magnetically"

_Annales Geophysicae, 2020_

## Referee Comment (RC1) · Anonymous Referee #1 · 5 Jul 2020

General comments: The authors conducted case studies to analyze the variations of >2 MeV electrons in terrestrial radiation belt, and their relation to interplanetary triggers and geomagnetic activities. However, neither any contribution can be found to further the current understanding of relativistic electron dynamics, nor the methods the authors applied are appropriate and valid.

Specific comments: 1. While the authors claimed that they extended the understanding of radiation belt dynamics, what has been shown in this manuscript is that they simply listed the previous findings of other researchers, without, throughout the manuscript, pointing out which aspects are still not clear and what they did to improve the under-

standing of these aspects. I think this is the main problem. This main flaw makes the current manuscript more an essay, rather than a research article.

2. The authors stated that this work is case study. Their method is to conduct cross correlation analyses between electron flux and different parameters, for each of the 4 cases, respectively. Then they drew conclusion on whether there is a relation between electron flux and the parameters. If the goal is to establish a link between different parameters, at least the cross correlation analysis should be applied to a large amount of events, so that the relation established is significant statistically. It is not acceptable to say one thing is related to another by cross correlation for only several cases.

3. I don't see the meaning of performing continuous wavelet transform in this manuscript. CWT is to discover periodicities or trends in time series——what trends in relativistic electron flux do the authors expect within the 2-3 hour time period of each event shown in Fig. 5? It seems that the authors blindly applied CWT to the current datasets, whereas they don't quite understand the fundamental mechanisms as well as the prerequisites of the methods they used.

Technical correction:

There are enormous typo errors throughout the manuscript, too many that it seems that the authors did not care about what they were writing.

---

## Referee Comment (RC2) · Anonymous Referee #2 · 13 Jul 2020

General comments:

The authors studied the relativistic electron flux (E > 2 MeV) in the outer radiation belt during four events, three magnetic storms with different intensities and a quiet period, using wavelet transform and cross-correlation. The solar wind parameters and a magnetic storm index have been related to the radiation belt electron flux. The case studies may be interesting although it should be carefully presented and explained. The manuscript is not clear in many parts and presents many language issues.

Specific comments:

1. Some parts in the manuscript are confusing as for instance the lines 17-18 and

300-301 do not agree in the statements.

2. There are strong conclusions during the manuscript mentioning previous papers although the results do not clearly show it.

3. The Introduction may be rewritten since it does not really support your work, mainly in lines 50-59. The Van Allen probes are mentioned in lines 50-51 but only GOES data is used.

4. How does your work focus on loss, acceleration and transport of relativistic electrons as mentioned in lines 61-63?

5. You mention that "magnetic storms are not the primary factor that pumps up the radiation belts", but you found a good correlation between electron flux and SYM-H. How do you explain that? Do you think your results support your conclusion?

6. What do you mean by "different interplanetary structures" in line 145? You only mention high speed streams.

7. What is the point of using Wavelet transform in your work to support your conclusions?

Technical corrections:

It has been pointed out some corrections, but not everything. You may please check punctuation, space, missing "the", references, etc.

15: remove and

15: fluctuation or variation?

16: is dependent

15-17: This conclusion is not clear during the manuscript.

17-18: This sentence does not agree with the statement in lines 300-301.

Interactive
comment

22: electron flux

22-24: The same comment is lines 334-336: You may be clear here that you are relating electron flux with SYM-H.

32-34: ...(CME), co-rotating interaction region (CIR) and high speed streams (HSS)

34: space before reference

34: add more references related to geomagnetic disturbances during CIR and HSS

36: trapping or loss of high

36: charged

36: particles in the Van Allen radiation belts (remove known as Van Allen belt)

37: space before reference

37-38: This sentence is not clear "There are . . . flux"

38-39: This sentence is not clear "Enhancement. . .atmosphere", add reference to it.

40-41: The sentence "Magnetic reconnection. . .magnetosphere." has no connection to the entire paragraph. I suggest removing it.

43: ions, protons? Would it be just ions?

44: The outer. . .

48: Replace drags us

43-59: I suggest to rewrite the second part of the paragraph: "The aftermath. . .. values.". You may explain some past results which are important to state your present work.

Section 2: Dataset and Methodology

61:loss, acceleration and transport?

65: dataset

67: Omni web link does not work.

95: Table

96-105: Missing space in tittle SYM-H value; SYM-H intervals may be rewritten, starting from the lower value to the higher (-50 to 0), choosing the word to or the inequality symbol, not both.

112-114: This sentence should be in the Methodology Section.

118: It is missing unit: -4 nT

119: which may be

121: Why to 2 to 2.5 nPa? At the plot the PSW reaches lower values.

121: solar wind pressure? Which pressure? Dynamic? Thermal? Magnetic?

124-126- Sentence "As the solar wind. . ." is not clear.

126-128- Sentence "Since. . ." is part of the last sentence. Both may be rewritten.

128: fluxes -> flux

131-132: Sentence "As high speed. . ." is not clear.

134: corresponding to the time of minimum SYM-H value.

145: What do you mean by "different interplanetary structures"? It may be clear in the sentence.

146-147: The sequence of panels is the same as explained in Figure 1.

147-148: which indicates the storm is moderate according to . . .. . .

148: remove [ ]

149 : allows the charged

157: You'd rather rewrite it since accelerate is not a good word here.

158: new paragraph

164: rewrite reference

167: stream

168: storm

168: The higher solar wind speed, the higher

182-183: Sentence "The fluctuation..." is not clear.

185: remove [ ]

186-188: Sentence may be rewritten.

190: accelerating is not appropriate here.

191: What is normal?

193: fluxes or flux?

195-205: This discussion should be improved.

215: compression of bow shock?

229-241: This discussion should be improved.

275: Figure 5 should be presented in the same order as the Section, from quiet to super-intense storm.

280: "(refer to ...)"is not clear.

286: "our work" may not be necessary.

288: for the intense storm

299: greater -> larger

301: The sentence "Hence .." may be rewritten.

314: events

314: there seems to increase?

320-321: "compressed . . .far" is not clear.

322: and enhancing

326: 'To be ..' is not clear

328: in all

328: the intense

329: The high Psw values lead

334-336: You may be clear here that you are relating electron flux with SYM-H.

336-339: it is not clear.

-You did not mention anything about the red dashed line in Figures 2, 3 and 4.

-The description of similar figures may follow the same pattern in all the figures and Sections.

-You refer to figures as Figures, figures, fig., etc. This should follow the same pattern along the manuscript.

- Replace solar wind velocity by solar wind speed.

---

## Author Comment (AC1) · 7 Aug 2020

We thank the referee for reviewing our manuscript and for providing detailed comments. The reviewer raised several major concerns regarding the techniques employed in our study, which we address in detail below. As can be seen from our response, we feel that these issues can be resolved by better clarifying our methodology in the revised version of our manuscript. Our responses are marked in red color font in the text below, in between the reviewer's comments.

General comments: The authors conducted case studies to analyze the variations of >2 MeV electrons in terrestrial radiation belt, and their relation to interplanetary triggers

[Figure]

and geomagnetic activities. However, neither any contribution can be found to further the current understanding of relativistic electron dynamics, nor the methods the authors applied are appropriate and valid. Specific comments: 1. While the authors claimed that they extended the understanding of radiation belt dynamics, what has been shown in this manuscript is that they simply listed the previous findings of other researchers, without, throughout the manuscript, pointing out which aspects are still not clear and what they did to improve the understanding of these aspects. I think this is the main problem. This main flaw makes the current manuscript more an essay, rather than a research article.

Response: The scope of the paper will be included in the revised manuscript:

The population of relativistic electron fluxes in the Earth's radiation belt are highly variable which are supposed to be balanced by a complicated and delicate balance between acceleration, transport, and loss processes. It is one of the subjects in space science of intense interest and study. There have been vigorous studies and research about those killer electrons (named since its adverse effects to astronauts and satellites onboard), still, there is a lack of understanding about their relation and periodicities with so-called geomagnetic storms. This casework thus tried to comprehend its periodic nature and dependency with geomagnetic storms using cross correlation and wavelet analysis. This paper addresses the trend and association of relativistic electron fluxes with geomagnetic storms taking important solar interplanetary parameters. The discussion and conclusion of this casework is believed to serve as a reference for the times to come in the field of space science.

In the revised manuscript, we will revise the introduction section and add two new events as below.

Introduction

The major plasma sources in the interplanetary medium responsible for geomagnetic disturbances are identified as coronal mass ejection (CMEs), which include the magnetic cloud, interplanetary shock, the co-rotating interaction region (CIR) and the high-speed solar wind streamers[Gosling et al., 1991]. The interaction between those interplanetary structures with the Earth's magnetosphere-ionosphere system can produce effects such as geomagnetic storms, sub storms and trapping of high energy charge particles in the radiation belt, known as Van Allen belt [Mauk et al., 2012]. There are various solar wind parameters that are effective enough to fluctuate the content of relativistic electron flux. Enhancement in relativistic electron fluxes might be an important sources of energy input and chemical change to the middle atmosphere. Magnetic reconnection is the main physical phenomena transporting energy from the solar wind into the magnetosphere.

Van Allen radiation belts are composed of ions, protons and electrons with energy ranging from 100 k eV to 10 MeV. It consists of two belts: inner and outer radiation belt. Outer radiation belt usually lies at an altitude of 3 Earth radii (RE) and extending to 10 RE above the Earth's surface where GPS satellites, metrological satellites, broadcasting, and communication satellites are operating [R. Kataoka and Y. Miyoshi, 2008]. The increasing dependency on modern infrastructure and technology and expanding human presence in space drags us more for the comprehensive study and understanding of space weather and their dynamics [Baker et al., 2000]. The variation of relativistic electron fluxes is supposed to be controlled by a delicate and complicated balance of acceleration and loss processes during storm conditions. The interplay between acceleration, loss and transport mechanism brings out the wide range of variability in radiation belts. Initially, the relativistic electrons were reported by Paulikas and Blake [1979] who observed that upon impinging the magnetosphere the electrons appeared were associate with high speed solar wind streamers [Hajra et al., 2013,2015].

The accepted standard theory for the source of outer belt relativistic electrons is a three step process: injection of substorm electrons($\sim$10-100 keV) followed by locally acceleration of $\sim$100 keV electrons by whistler-mode chorus wave and then finally redistribution by radial diffusion [ Meredith et al., 2003; Miyoshi et al., 2003; Thorne et

al., 2013; Li et al., 2014]. For the deeper understanding of the structure and dynamics of Earth's radiation belt, NASA developed the Van Allen Probes mission [Mauk et al., 2012].The aftermath of highly fluctuating electron fluxes in the Earth's outer radiation belt might be its acceleration and loss. Paulikas & Blake [1979] reported such rapid acceleration and loss of relativistic electrons. Reeves et al., [2003] and Turner et al. [2013] added relativistic electron population in the radiation belt can not only subsidize but also can be enhanced, depleted, or even not affected at all due to the acceleration and loss mechanism. Pinto et al. [2018] identified and analyzed 61 relativistic electron enhancement events and 21 depletion events during 1996 to 2006, resulting the persistent depletion events are characterized by: a low Vsw, a sudden increase in proton density, and a northward turning of IMF-Bz. Also, predicted their threshold values.

The population of relativistic electron fluxes in the Earth's radiation belt are highly variable which are supposed to be balanced by a complicated and delicate balance between acceleration, transport, and loss processes [Reeves et al., 2003; Turner et al., 2013]. It is one of the subjects in space science of intense interest and study. There have been vigorous studies and research about those killer electrons (named since its adverse effects to astronauts and satellites onboard), still, there is a lack of understanding about their relation and periodicities with so-called geomagnetic storms. This casework thus tried to comprehend its periodic nature and dependency with moderate storm, intense storm, super-intense storm, storm and non-storm HILDCAAs (high intensity long duration continuous aurora activity) and one geo-magnetically quiet period using cross correlation and wavelet analysis. This paper addresses the trend and association of relativistic electron fluxes with geomagnetic storms taking important solar interplanetary parameters.

An ICME preceding HILDCAA during 15-18 May 2005 Figure 4a shows HILDCAA preceded by ICME on 15-18 May 2005. The panel of the plot is the same as described in previous Figures. The interplanetary cause of this storm was the shock driven by an ICME containing a magnetic cloud structure [Hajra et al., 2013; Ojeda et al., 2013]

characterized by large southward IMF-Bz with a peak of −20 nT. The storm main phase starts at ∼8 hrs. During the storm onset time, the ICMEs are faster enough (solar wind speed Vsw>800 km/s) to form a forward shock [Kennel, 1985]. These interplanetary structures contain relatively high density (Nsw∼30 n/cc) and solar dynamic pressure (Psw∼45 nPa) compared to the normal solar wind. The interaction of these structures with the front of the magnetosphere causes compression of the magnetosphere and can cause magnetopause shadowing losses [Nishida, 1978; Kim et al., 2010; Hietala et al., 2014]. This is depicted in the plot as the relativistic flux is very low and constant for almost a day. The corresponding SYM-H value is -300 nT with AE ∼1000 nT. The High-Intensity, Long-Duration, Continuous AE Activity, or HILDCAA event starts with the recovery phase after mid-day of 15 May for which the AE value ∼1000 nT [Tsurutani and Gonzalez, 1987]. During HILDCAA, the IMF Bz directed towards southwards (Bz ∼-20 nT and remains modest for the rest of the day), solar wind speed gradually decreases and so does the pressure and density. After nearly 1-day of HILDCAA, the flux of relativistic electrons starts to accelerate ($\sim 10 \times 10^4$ FU). This intensification of flux maintained for the whole HILDCAA event. This is in agreement with the study of Guarnieri et al. [2006], who noted that the HILDCAA may accelerate the flux of relativistic electrons. References: Hietala, H., E. K. J. Kilpua, D. L. Turner, and V. Angelopoulos (2014), Depleting effects of ICME-driven sheath regions on the outer electron radiation belt, Geophys. Res. Lett., 41, 2258–2265, doi:10.1002/2014GL059551 Nishida, A. (1978), Geomagnetic Diagnosis of the Magnetosphere (Physics and Chemistry in Space). [S.l.]: Springer-Verlag; First Edition edition. Hajra, R.; Echer, E.; Tsurutani, B. T.; Gonzalez, W. D(2013), Solar cycle dependence of high-intensity, long-duration, continuous ae activity (hildcaa) events. Journal of Geophysical Research,V-s118.

Ojeda, G. A.; Mendes, O.; Calzadilla, M. A.; Domingues, M. O(2013), Spatio–temporal entropy analysis of the magnetic field to help magnetic cloud characterization, Journal of Geophysical Research, V-118, p. 5403–5414.

Non-storm HILDCAA during 20-23 April 2003 Figure 4b represents the signatures of

the Non-storm HILDCAA event of 20-23 April 2003 which is not preceded by the geomagnetic storm. The IMF Bz fluctuations were around zero with amplitudes around $\pm 8$ nT. The solar wind speed Vsw remains fairly constant with an average value of $\sim 560$ km/s. The SYM-H value drops to $\sim$-40 nT at 25 hrs and exhibits one decrease (-60 nT at 80 hrs). The corresponding AE value was $\sim 1100$ nT, indicating HILDCAA event. The fluctuations of solar dynamic pressure and solar density were fairly similar and low. During the event, the flux of the relativistic electron has an average value of $\sim 0.5 \times 104$ FU until $\sim 62$ hrs and then increases abruptly with $\sim 1..8 \times 104$ FU for nearly 16 hrs and again decreases to normal value. Although this event is characterized by high auroral activity via particle injection but does not clearly indicate either of the acceleration or loss of energetic particles. Figure 4a: From top to bottom, the panels show the variations of: solar wind pressure(Psw in nPa), solar wind speed (Vsw in km/s), interplanetary magnetic field (Bz in nT), solar wind plasma density(Nsw in n/cc), relativistic electron flux ðİŘÿ > 2 ðİŠÅðİŠŠðİŠĽ(Flux in ðİŠŘðİŠŽ−2ðİŠĘ −1ðİŠĘðİŠ§ −1 ), symmetric horizontal component of magnetic field (SYM-H in nT), and auroral electrojet (AE in nT) indices with time (Hours) respectively for event-4 of 15-18 May 2005. Figure 4b: From top to bottom, the panels show the variations of: solar wind pressure(Psw in nPa), solar wind speed (Vsw in km/s), interplanetary magnetic field (Bz in nT), solar wind plasma density(Nsw in n/cc), relativistic electron flux ðİŘÿ > 2 ðİŠÅðİŠŠðİŠĽ(Flux in ðİŠŘðİŠŽ−2ðİŠĘ −1ðİŠĘðİŠ§ −1 ), symmetric horizontal component of magnetic field (SYM-H in nT), and auroral electrojet (AE in nT) indices with time (Hours) respectively for event-4 of 20-23 April 2003.

2. The authors stated that this work is case study. Their method is to conduct cross correlation analyses between electron flux and different parameters, for each of the 4 cases, respectively. Then they drew conclusion on whether there is a relation between electron flux and the parameters. If the goal is to establish a link between different parameters, at least the cross-correlation analysis should be applied to a large amount of events, so that the relation established is significant statistically. It is not acceptable to say one thing is related to another by cross correlation for only several cases.

[Figure]

3. I do not see the meaning of performing continuous wavelet transform in this manuscript. CWT is to discover periodicities or trends in time series'ⅇ AĚŸTwhat trends ĚĞ in relativistic electron flux do the authors expect within the 2-3-hour time period of each event shown in Fig. 5? It seems that the authors blindly applied CWT to the current datasets, whereas they don't quite understand the fundamental mechanisms as well as the prerequisites of the methods they used.

Response:

Figures 7 (a-b) show the relationship of relativistic electrons with solar wind parameters and geomagnetic indices for the time interval of respective six events: super intense storm, ICME-HILDCAA, Intense storm, Moderate storm, non-storm HILDCAA, and geomagnetically Quiet event. The use of wavelet analysis for the signal of relativistic electrons helped us to reduce the unnecessary signal for the proper understanding of their relationship with solar wind parameters and geomagnetic indices using cross-correlation techniques. Unlike the event of 20-23 April 2003 showing a moderate correlation of Relativistic electrons with Vsw, Psw, and NSW; the other five events depict linearly good correlation with cross-correlation coefficient of greater than 0.7 at zero time lags. However, it shows strong anti-correlation with SYM-H value. The highly fluctuating line of Rel-Bz in the event of 25th January 2007 shows a lesser cross-correlation coefficient between them [Adhikari et al, 2017]. That means the response of the southward flow of the magnetic field shows a mediocre response towards the signal of relativistic electrons even when the plasma pressure and solar wind velocity are in good agreement with the relativistic electrons. Besides, IMF-Bz with Relativistic electrons shows anti-correlation in the other five events. The higher correlation shows the linear relationship between them which suggests the enhancement of relativistic electrons while decreasing relationship shows the loss or no response during different events. Table (2) depicts the enhancement or depletion of relativistic electrons during the time of different events discussed in this study [Anderson et al., 2015]. Events of 3-4 September 2008 and 30-31 March 2001 with ratio (post-storm to pre-storm) value

of 3.86 and 4.78 respectively indicate the enhancement of relativistic electrons during these event intervals; however, depletion of relativistic electrons is experienced on other events. No change in relativistic electrons is not experienced. Events Relativistic electrons ratios (Poststorm/Prestorm) 20-23 April 2003(Non-storm HILDCAA) 1.70559 15-18 May 2005 (ICME- HILDCAA ) 1.6798 3-4 September 2008 (Moderate) 3.86280 14-15 December 2006 (Intense) 1.674387 30-31 March 2001 (sss) 4.7852169 24-25 January 2007 (quiet) 0.685581 Table 2: Shows the enhancement or no change and depletion of relativistic electrons during different events of geomagnetic storms.

Figure 7 (a): Cross-correlation between Relativistic electron flux (E>2MeV) with Bz (blue), Vsw (green), Psw (red), SYM-H (sky blue) and Nsw (pink) during 20th-223rd April 2003.

Figure 7 (b): Cross-correlation between Relativistic electron flux (E>2MeV) with Bz (blue), Vsw (green), Psw (red), SYM-H (sky blue) and Nsw (pink) during 15th May 2005.

References: Adhikari, Binod, Prashrit Baruwal, and Narayan P. Chapagain. "Analysis of supersubstorm events with reference to polar cap potential and polar cap index." Earth and Space Science 4.1 (2017): 2-15. Anderson, B. R., R. M. Millan, G. D. Reeves, and R. H. W. Friedel (2015), Acceleration and loss of relativistic electrons during small geomagnetic storms, Geophys. Res. Lett., 42, 10,113–10,119, doi:10.1002/2015GL066376.

Wavelet: Power spectrum and global wavelet spectrum In order to substantiate the obtained results and for the close inspection of the existing trends of relativistic electrons in six time-series events, the use of wavelet: wavelet power spectrum (WPS) and global wavelet spectrum (GWS) techniques are adopted. These techniques provide an unbiased and true estimation of periodicity as the original signal gets decomposed to several components using continuous wavelet transform (CWT) [Torrence et al., 1998; Markovic et al., 2005; Santos et al., 2013]. The upper panel provides the original data

of relativistic electrons of which wavelet is applied. The bottom panel is the scale-ogram or squared modulus of the wavelet coefficients of CWT, suggesting the energy distribution over time series. The Y-axis represents the scale or the period in minutes (0.0625, 0.125, 0.5, 1, 2, 4, etc.) that depicts the oscillations of the signal within the individual wavelet concerning the time series in Hours plotted on X-axis [Khanal et al., 2019]. Also, Scale = 1/ frequency such that, lower the scale value higher is the frequency and vice versa. That the thin black line curve, as a cone of influence, divides the time-frequency system into two parts which show the validity of data within the curve; however, limits the usability outside the curve. Moreover, the appearance of color index intensity circumvents with black contour indicates the concentration of power with 95% confidence level or a 5% level of significance [Markovic et al., 2005; Manyilizu et al., 2014; Yan et al., 2017]. The color index represents the intensity of relativistic electrons. The peak on the GWS resembles the higher color index value [Santos et al., 2005]. Santos, et al. (2001) suggested that the accumulated power of GWS occurs, particularly, between high frequencies. However, Falayi et al., (2020) obtained the concentration of power between larger period bands across solar wind parameters and geomagnetic indices. This study suggests that the oscillation and the concentration of power of relativistic electrons showed significant peaks across the low-frequency periods (e.g. 1, 2 and 4 minutes) in six events: Super intense storm, ICME-HILDCAA, Intense storm, Moderate, Non-HILDCAA, and Quiet events; however, indicate that these responses are usual characteristics in the WPS and GWS analysis. Furthermore, the integrated power generated using GWS of the signals of relativistic electrons might be unreliable for the circumstantial factors for the identification of solar activity since the geomagnetic storms are the aftermaths of the solar activities and relativistic electrons emanating from such activities do not exhibit significant role during the main events of geomagnetic storms. Figures from 8 -13 show the wavelet analysis of Relativistic electrons during the time interval of geomagnetic storms. The upper panels show the Relativistic electron obtained with the goes 10 satellite in a real-time scenario. The bottom panel indicates the wavelet spectrum. The colors index represents the intensity of the power spectrum. The right panel plot supports the result of a scaleogram showing the total power concentrated for periods and regions of events.

Figure 8: Represents the wavelet: Power spectrum and Global wavelet spectrum analysis of the relativistic electron during the super intense storm for the interval shown in Figure of 30-31 March 2001. The wavelet of relativistic electrons is of the order of 10ˆ3 in an event of 30-31 March 2001 as a super intense storm is analyzed using WPS (or scaleogram) and GWS shown in Figure 7. The gradual increment in the intensity– the distribution of color index at time intervals for 18th -26th hours – from low period to higher period in minutes confirms the occurrence of the main event with an accumulated power, dashed black line on GWS, of approx. $3.5\times10ˆ4$ which depicts the flow of relativistic electrons in the magnetosphere-ionosphere system (MIS). Furthermore, the dark black contour at 22nd hours of scaleogram shows the 95% confidence interval at which the maximum power is experienced on the MIS. The power spectrum helps to identify the impacts of the relativistic electrons during the main event. No significant effect of Relativistic electrons are experienced both prior to the 16th hours and after the 26th hours for the event of 30-31 March 2001. A power greater than on average $2\times10ˆ4$ is observed for 1-2 minutes for nearly 5th hours. The increment of power, unbiased and consistent, in GWS, hints the most turbulent times that might represent the significant impact on space weather. Figure 8 depicts the power (as an absolute value squared) generated using GWS of the wavelet transform for relativistic electrons during the time interval of a super intense storm. The (absolute value)ˆ2 gives information on the relative power at a certain scale and a certain time. Furthermore, there is more concentration of power between the 0.5 -2 minutes band, which shows that this time series has a strong hours signal. The wavelet power spectrum for relativistic electrons episodes with a characteristic scale of 1-2 minutes presents an important peak. Using wavelet analysis, we show that the oscillation and accumulation of power for relativistic electrons dominated the low-frequency region (i.e., on the higher period (minutes)).

Figure 9: Represents the wavelet: Power spectrum and Global wavelet spectrum analysis of the relativistic electron during ICME-HILDCAA for the interval shown Figure of 15-18 May 2005.

We can identify the discrete and continuous oscillations during various events (ICME-HILDCAA, Super Intense storm, moderate storm, non-storm HILDCAA, and geomagnetically Quiet storm) with peaks consistent in time and period resembling the injection of relativistic electrons in the magnetosphere. In this event, the GWS detects multiple peaks of power having values 2.5 and $5.2 \times 10^9$ at 1.5 and 3 periods, respectively. The broader distribution of color is observed at around 3 periods from 50th to 86th hours i.e. nearly 36th hours. The recovery phase lasted for 20 hours. However, the level of significance is not in correlation with the power curve. The SYM/H value prior to 40th hours in Figure 4b suggests the main event of ICME-HILDCAA; however, no significant effect of relativistic electrons is observed. Which is in agreement with the findings of Hajra et al., (2015). The effects at the 70th -90th hours are due to the perturbation of relativistic electrons during the recovery phase. Unlike other events, a large accumulation of power is observed during the recovery phase of ICME-HILDCAA.

Figure 10: Represents the wavelet: Power spectrum and Global wavelet spectrum analysis of the relativistic electron during the intense storm for the interval shown in Figure of 15th December 2006.

In Figure 10, the distribution of color is observed from 11th -18th hours. The presence of relativistic electrons is observed at the 14th hour. The effect of a relativistic electron shows no prior effects before 12th hours. The distribution color greater than 1.5 clarifies that the event process remained for 6th hours, while the maximum power of $\sim 1.5 \times 10^8$ is observed at 14:30th Hours at a period (minutes) of 1-2. Short discontinuous signals are observed at lower period regions. Yet, the oscillation of signals is more continuous and broader at higher periods (minutes). WPS confirms and reveals an appropriate perturbed state at a variety of periods in correspondence to the temporal variability in Relativistic electrons. Furthermore, the GWS depicts the integrated maximum power accumulation during the main event due to the oscillatory activity of

relativistic electrons.

Figure 11: Represents the wavelet: Power spectrum and Global wavelet spectrum analysis of the relativistic electron during the moderate storm for the interval shown in Figure of 4th September 2008. In figure 11 the distribution of color index is observed only after 12th hours, even though the order of relativistic electrons is 10ˆ4. At 16th hours' time interval the variation of intensity shows that the gradual oscillations from 0.125 till 1 are experienced. The low power value is seen unlike other events including the geo-magnetically quiet event. This suggests that the value of relativistic electrons is not only the factor for change in power generated in the GWS.

Figure 12: Represents the wavelet: Power spectrum and Global wavelet spectrum analysis of the relativistic electron during non-storm HILDCAA for the interval shown in Figure of 20-23 April 2003.

In figure 12 the time-series signal of relativistic electrons during Non-storm HILDCAA (20-23 April 2003) shows the presence of a 95% level of significance indicated by the dark contour for a few hours from 65th to 75th hours interval showing oscillations within scales from 1.5 to 3. Besides, the peak of power was observed in the order of 10E6. No oscillations for low periods are observed.

Figure 13: Represents the wavelet: Power spectrum and Global wavelet spectrum analysis of the relativistic electron during geomagnetically quiet for the interval shown in Figure 3-4 September 2008. The smooth and periodic trend of relativistic electrons during geomagnetically quiet periods in Figure 13 of 3-4 September 2008 was observed. No intense color distribution is seen. The period provides oscillations within the wavelet. The relativistic electrons number is of the order one; however, the GWS shows the power to the order of 10ˆ5 which is comparable to the intense storm and greater than power during the moderate storm which is of the order 10ˆ1. These suggest that the curve of power hinges on the uniform and periodic flow of the relativistic electrons on space weather. References: Santos, Celso Augusto Guimarães, and

Bruno Sousa de Morais. "Identification of precipitation zones within São Francisco River basin (Brazil) by global wavelet power spectra." Hydrological sciences journal 58.4 (2013): 789-796.

Torrence, Christopher, and Gilbert P. Compo. "A practical guide to wavelet analysis." Bulletin of the American Meteorological society 79.1 (1998): 61-78.

Markovic, D., and M. Koch. "Wavelet and scaling analysis of monthly precipitation extremes in Germany in the 20th century: Interannual to interdecadal oscillations and the North Atlantic Oscillation influence." Water Resources Research 41.9 (2005).

Yan, R., Woith, H., Wang, R. et al. Decadal radon cycles in a hot spring. Sci Rep 7, 12120 (2017). https://doi.org/10.1038/s41598-017-12441-0.

SANTOS, Celso AG, et al. "Matsuyama city rainfall data analysis using wavelet transform." PROCEEDINGS OF HYDRAULIC ENGINEERING 45 (2001): 211-216. Khanal, Krishna, et al. "HILDCAA‐related GIC and possible corrosion Hazard in underground pipelines: A comparison based on wavelet transform." Space Weather 17.2 (2019): 238-251.

Falayi, E. O., et al. "Study of nonlinear time for greater analysis and wavelet power spectrum analysis using solar wind parameters and geomagnetic indices." NRIAG Journal of Astronomy and Geophysics 9.1 (2020): 226-237.

Manyilizu, M., et al. "Interannual variability of sea surface temperature and circulation in the tropical western Indian Ocean." African Journal of Marine Science 36.2 (2014): 233-252. Hajra, R., Tsurutani, B.T., Echer, E. et al. Relativistic electron acceleration during HILDCAA events: are precursor CIR magnetic storms important?. Earth Planet Sp 67, 109 (2015). https://doi.org/10.1186/s40623-015-0280-5.

Technical correction: There are enormous typo errors throughout the manuscript, too many that it seems that the authors did not care about what they were writing.

Response: We are not sure where the reviewer refers to this comment, in the revised

manuscript, we will remove all typo errors and provide the manuscript with cogent analysis for greater clarity.

[Figure]

[Figure]

**Fig. 1.**

[Figure]

**Fig. 2.**

[Figure]

**20th-23rd April 2003**

| Legend |
|---|
| Rel-Bz |
| Rel-Vsw |
| Rel-Psw |
| Rel-SYM(H) |
| Rel-Nsw |

**Fig. 3.**

[Figure]

**Fig. 4.**

[Figure]

**Fig. 5.**

[Figure]

**Fig. 6.**

[Figure]

Fig. 7.

[Figure]

**Fig. 8.**

[Figure]

**Fig. 9.**

**Fig. 10.**

| Events | Relativistic electrons ratios (Poststorm/Prestorm) |
|---|---|
| 20-23 April 2003(Non-storm HILDCAA) | 1.70559 |
| 15-18 May 2005 (ICME- HILDCAA ) | 1.6798 |
| 3-4 September 2008 (Moderate) | 3.86280 |
| 14-15 December 2006 (Intense) | 1.674387 |
| 30-31 March 2001 (sss) | 4.7852169 |
| 24-25 January 2007 (quiet) | 0.685581 |

**Fig. 11.**

---

## Author Comment (AC2) · 7 Aug 2020

We thank the referee for reviewing our manuscript and for providing positive and detailed suggestions. Our responses are marked in the "red color" font in the text below, in between the reviewer's comments.

General comments: The authors studied the relativistic electron flux (E > 2 MeV) in the outer radiation belt during four events, three magnetic storms with different intensities and a quiet period, using wavelet transform and cross-correlation. The solar wind parameters and a magnetic storm index have been related to the radiation belt electron flux. The case studies may be interesting although it should be carefully presented and

[Figure]

explained. The manuscript is not clear in many parts and presents many language issues.

Specific comments: 1. Some parts in the manuscript are confusing as for instance the lines 17-18 and 300-301 do not agree in the statements.

Response: In the revised manuscript, a sentence of Line 17-18 will be replaced with "cross-correlation analysis depicted that the response of relativistic electrons with SYM-H showed good anti-correlation".

2. There are strong conclusions during the manuscript mentioning previous papers although the results do not clearly show it.

Response: We are not sure where the reviewer refers to this comment; however, the additional analysis presented in the revised manuscript might have resolved such issues.

3. The Introduction may be rewritten since it does not really support your work, mainly in lines 50-59. The Van Allen probes are mentioned in lines 50-51 but only GOES data is used.

Response: Thanks for pointing out the problems. The introduction part is revised and will be included in the revised manuscript. Also included in the reviewer's 1 comment.

4. How does your work focus on loss, acceleration and transport of relativistic electrons as mentioned in lines 61-63?

Response: To resolve this issue, we have included additional analysis with the necessary data in the table (2) which will be included in the revised manuscript. (included in a reviewer 1's comment file).

5. You mention that "magnetic storms are not the primary factor that pumps up the radiation belts", but you found a good correlation between electron flux and SYM-H. How do you explain that? Do you think your results support your conclusion?

Response: Thanks for pointing out flaws. This statement is rewritten in the revised manuscript and in accordance with the conclusion.

6. What do you mean by "different interplanetary structures" in line 145? You only mention high speed streams.

Response: Geomagnetic storms are the consequences of different interplanetary structures like coronal mass ejections, interplanetary coronal mass ejections, high-speed streams, etc. This sentence will be resolved in the revised manuscript.

7. What is the point of using Wavelet transform in your work to support your conclusions?

Response: In order to substantiate the obtained results and for the close inspection of the existing trends of relativistic electrons in six time-series events, the use of wavelet: wavelet power spectrum (WPS) and global wavelet spectrum (GWS) techniques are adopted. These techniques provide an unbiased and true estimation of periodicity as the original signal gets decomposed to several components using continuous wavelet transform (CWT).

Technical corrections: It has been pointed out some corrections, but not everything. You may please check punctuation, space, missing "the", references, etc. 15: remove and 15: fluctuation or variation? 16: is dependent 15-17: This conclusion is not clear during the manuscript. 17-18: This sentence does not agree with the statement in lines 300-301. 22: electron flux 22-24: The same comment is lines 334-336: You may be clear here that you are relating electron flux with SYM-H. 32-34: . . .(CME), co-rotating interaction region (CIR) and high speed streams (HSS) 34: space before reference 34: add more references related to geomagnetic disturbances during CIR and HSS 36: trapping or loss of high 36: charged 36: particles in the Van Allen radiation belts (remove known as Van Allen belt) 37: space before reference 37-38: This sentence is not clear "There are . . . flux" 38-39: This sentence is not clear "Enhancement. . .atmosphere", add reference to it. 40-41: The sentence "Magnetic reconnection. .

.magnetosphere." has no connection to the entire paragraph. I suggest removing it. 43: ions, protons? Would it be just ions? 44: The outer. . . 48: Replace drags us 43-59: I suggest to rewrite the second part of the paragraph: "The aftermath. . .. values.". You may explain some past results which are important to state your present work.

Section 2: Dataset and Methodology 61:loss, acceleration and transport? 65: dataset 67: Omni web link does not work. Response: link will be replaced as https://omniweb.gsfc.nasa.gov in the revised manuscript. 95: Table 96-105: Missing space in tittle SYM-H value; SYM-H intervals may be rewritten, starting from the lower value to the higher (-50 to 0), choosing the word to or the inequality symbol, not both. 112-114: This sentence should be in the Methodology Section. 118: It is missing unit: -4 nT 119: which may be 121: Why to 2 to 2.5 nPa? At the plot the PSW reaches lower values. 121: solar wind pressure? Which pressure? Dynamic? Thermal? Magnetic? 124-126- Sentence "As the solar wind. . ." is not clear. 126-128- Sentence "Since. . ." is part of the last sentence. Both may be rewritten. 128: fluxes -> flux 131-132: Sentence "As high speed. . ." is not clear. 134: corresponding to the time of minimum SYM-H value. 145: What do you mean by "different interplanetary structures"? It may be clear in the sentence. 146-147: The sequence of panels is the same as explained in Figure 1. 147-148: which indicates the storm is moderate according to . . .. . . 148: remove [ ] 149 : allows the charged 157: You'd rather rewrite it since accelerate is not a good word here. 158: new paragraph 164: rewrite reference 167: stream 168: storm 168: The higher solar wind speed, the higher 182-183: Sentence "The fluctuation. . ." is not clear. 185: remove [ ] 186-188: Sentence may be rewritten. 190: accelerating is not appropriate here. 191: What is normal? 193: fluxes or flux? 195-205: This discussion should be improved. 215: compression of bow shock? 229-241: This discussion should be improved. 275: Figure 5 should be presented in the same order as the Section, from quiet to super-intense storm. 280: "(refer to . . .)"is not clear. 286: "our work" may not be necessary. 288: for the intense storm 299: greater -> larger 301: The sentence "Hence .." may be rewritten. 314: events 314: there seems to increase? 320-321: "compressed . . .far" is not clear. 322: and enhancing 326: 'To

be .." is not clear 328: in all 328: the intense 329: The high Psw values lead 334-336: You may be clear here that you are relating electron flux with SYM-H. 336-339: it is not clear. -You did not mention anything about the red dashed line in Figures 2, 3 and 4. -The description of similar figures may follow the same pattern in all the figures and Sections. -You refer to figures as Figures, figures, fig., etc. This should follow the same pattern along the manuscript. - Replace solar wind velocity by solar wind speed.

Thank you for your meticulous details. As suggested by the reviewer we will address all the stated comments and will be placed in the revised manuscript.